# Probability-Box Informed Hysteresis Modelling through Metaheuristic Search Algorithms

**Jone Ugarte-Valdivielso**
Department of Electronics & Computing Science
Mondragon Unibertsitatea
jugarte@mondragon.edu

**Jose I. Aizpurua**
Department Computer Science and Artificial Intelligence, University of the Basque Country (UPV/EHU)
Ikerbasque, Basque Foundation for Science
joxe.aizpurua@ehu.eus

**Manex Barrenetxea**
Department of Electronics & Computing Science
Mondragon Unibertsitatea
mbarrenetxeai@mondragon.edu

## Abstract

Hysteresis modelling is crucial for many industrial applications ranging from material science to power and electrical energy systems. A frequently used approach in the magnetic materials and employed in the power and energy sector is the Jiles-Atherton (JA) model, which approximates the hysteresis curve through a partial differential equation (PDE). However, the parameter-estimation for the PDE is challenging. The present study evaluates the JA parameter estimation through the integration of probability-box (p-box) parameter initialization with metaheuristic search algorithms. The proposed p-box informed parameter initialization is tested for two different iron core materials integrated with three different metaheuristic-search algorithms, including Genetic Algorithms (GA), Particle Swarm Optimization (PSO) and Differential Evolution (DE). Then, the p-box approach is compared against the classical uniform and normal distribution based parameter initialization strategies. The results show that p-box parameter initialization can be used to estimate JA parameters accurately when there is little knowledge about the magnetic material and the transformer.

## 1 Introduction

Magnetic hysteresis occurs when a magnetic field is applied to a ferromagnetic material such as iron. Even if the field is removed, the material remains magnetized because of the alignment of its magnetic domains [1]. This makes it difficult to determine the exact magnetization state of the materials. Therefore, when an external field is applied, high magnetizing currents known as inrush currents can be generated. These currents appear due to the magnetic saturation of the iron core described by the $BH$ or hysteresis curve. An example of a hysteresis curve is shown in Figure 1, which corresponds to the H75-23 magnetic material used in medium-voltage power transformers.

An accurate model of the core is essential to reproduce the transient phenomena accurately. In this context, Jiles-Atherton (JA) is a widely used method for hysteresis modelling and inrush current

XVI XVI Congreso Español de Metaheurísticas, Algoritmos Evolutivos y Bioinspirados (maeb 2025).

minimization studies [2]. The JA model solves a partial differential equation (PDE) through an equivalent method. This model requires five parameters, whose accurate estimation is challenging. The simplest technique to estimate the JA parameters is a trial-and-error brute force process [3]. However, alternative search techniques, such as metaheuristic-based algorithms, are preferred due to their shorter computational time and efficient search for the global minimum error. In this context, the most used metaheuristic-based search techniques are Genetic Algorithm (GA) [4], Particle Swarm Optimization (PSO) [5], and Differential Evolution (DE) [6].

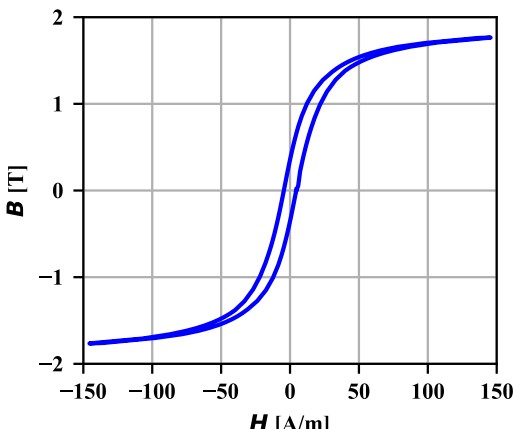

Figure 1: $BH$ curve for the H75-23 magnetic core material.

The selection of appropriate initial values and search limits for JA parameter estimation with metaheuristic-based algorithms is crucial, as it can avoid issues such as convergence to a local minimum, non-convergence, or high computational time. However, most of the presented studies in hysteresis modelling disregard the initialization stage and use a random JA parameter initialization strategy with default settings [7]. Moreover, probability density function (PDF) based parameter initialization strategies can improve the accuracy of metaheuristic-based algorithms [8]. Previous studies incorporating PDFs into the initialization of the JA parameter show enhanced accuracy and computational efficiency over random initialization strategies [9].

The JA parameters are related to the physical properties of the core material. The value of these parameters can be approximated using previous data and based on expert knowledge. However, this approximation can be subject to different levels of uncertainty depending on the reliability of the information and the experience and confidence of the expert. Furthermore, for diverse initialization strategies, e.g. various experts with different confidence and field knowledge, the approximations could be different. Therefore, to obtain a good approximation of the parameters considering all the available knowledge altogether, the different perspectives and the associated uncertainty levels should be considered. In this context, p-boxes present an interesting alternative to fuse different information criteria (modelling different initialization strategies) through imprecise probability concepts [10]. The uncertainty present in the JA parameter initialization can be classified into epistemic (e.g. the level of confidence of the expert at the parameter initialization stage), and aleatoric (e.g. uncertainty in the available magnetic material data) uncertainty [11]. In this regard, p-boxes are based on the combination of different cumulative distribution functions (CDFs) to quantify aleatoric and epistemic uncertainties. P-boxes can help with uncertain decision-making scenarios when different sources of information are available. Namely, different CDFs represent different pieces of knowledge of the process under consideration.

**Contribution**. This research work presents an original uncertainty-aware parameter initialization strategy for metaheuristic-search algorithms based on p-boxes. The proposed approach is evaluated through a detailed performance assessment framework used to estimate JA parameters for hysteresis modelling. Available data and expert knowledge are modelled and fused as p-boxes to improve parameter initialization and propagation. Obtained results are compared with the classical random uniform and normal PDF parameter initialization approaches for two different iron core materials.

## 2 Fundamental Theory

### 2.1 Basics of Jiles-Atherton

The Jiles-Atherton hysteresis approach models the total magnetization of a material by calculating and adding the reversible and the irreversible magnetization contributions defined as follows:

$$M = M_{\text{rev}} + M_{\text{irr}} \tag{1}$$

where $M_{\text{rev}}$ and $M_{\text{irr}}$ are, respectively, the reversible and irreversible terms in [A/m]. The irreversible magnetization is obtained as:

$$M_{\text{irr}} = M_{\text{an}} - k\delta\frac{\text{d}M_{\text{irr}}}{\text{d}H_{\text{e}}} \tag{2}$$

where $M_{\text{an}}$ is the anhysteretic magnetization curve in [A/m], $k$ is the pinning parameter in [A/m], $\delta$ is an indicator function that takes the value of +1 if $\text{d}H/\text{d}t > 0$ and -1 if $\text{d}H/\text{d}t < 0$, and $H_{\text{e}}$ is the effective field in [A/m] obtained from:

$$H_{\text{e}} = H + \alpha M \tag{3}$$

where $\alpha$ is unitless and refers to the interdomain coupling. The reversible magnetization is given as proportional to the difference between anhysteretic and hysteresis magnetization defined as follows:

$$M_{\text{rev}} = c(M_{\text{an}} - M_{\text{irr}}) \tag{4}$$

where $c$ is unitless and corresponds to the coefficient of proportionality. The anhysteretic magnetization curve can be expressed as:

$$M_{\text{an}} = M_{\text{s}}\left[\coth\left(\frac{H_{\text{e}}}{a}\right) - \frac{a}{H_{\text{e}}}\right] \tag{5}$$

where $M_{\text{s}}$ is the saturation magnetization in [A/m] and $a$ corresponds to the density of the wall in [A/m]. By combining all the previous equations, the JA model can be solved by applying the next PDE that corresponds to the magnetization susceptibility [12]:

$$\frac{\text{d}M}{\text{d}H} = \frac{(1-c)\frac{\text{d}M_{\text{irr}}}{\text{d}H_{\text{e}}} + c\frac{\text{d}M_{\text{an}}}{\text{d}H_{\text{e}}}}{1 - \alpha c\frac{\text{d}M_{\text{an}}}{\text{d}H_{\text{e}}} - \alpha(1-c)\frac{\text{d}M_{\text{irr}}}{\text{d}H_{\text{e}}}} \tag{6}$$

where $\frac{\text{d}M_{\text{irr}}}{\text{d}H_{\text{e}}}$ is expressed as follows:

$$\frac{\text{d}M_{\text{irr}}}{\text{d}H_{\text{e}}} = \frac{(M_{\text{an}} - M_{\text{irr}})}{k\delta} \tag{7}$$

Additionally, $\frac{\text{d}M_{\text{an}}}{\text{d}H_{\text{e}}}$ is obtained by deriving $M_{\text{an}}$ with respect to $H_{\text{e}}$ and is given as:

$$\frac{\text{d}M_{\text{an}}}{\text{d}H_{\text{e}}} = \frac{M_{\text{s}}}{a}\left[1 - \coth^2\left(\frac{H_{\text{e}}}{a}\right) - \left(\frac{a}{H_{\text{e}}}\right)^2\right] \tag{8}$$

The $MH$ curve is solved by calculating the magnetization value in each time step $\Delta t$ of $H$:

$$M(t + \Delta t) = M(t) + \frac{\text{d}M}{\text{d}H}\Delta H \tag{9}$$

where $t$ is the time step, $\Delta t$ is a discrete increase of $t$, $M(t + \Delta t)$ is the magnetization value at the instant $t + \Delta t$ in [A/m], $M(t)$ is the magnetization at $t$ in [A/m] and $\Delta H$ is the difference in the discrete increase of the field strength in [A/m].

From a practical point of view, it is preferable to give the results of the JA method in $BH$ instead of $MH$. The relation between $M$ and $B$ is established following the *Sommerfeld* convention. This equation is expressed as [1]: $B = \mu_0(H + M)$, where $\mu_0$ is the permeability of the vacuum with a value of $4\pi10^{-7}$ [H/m]. The flux density in each time step is obtained as follows:

$$B(t + \Delta t) = \mu_0\left[H(t + \Delta t) + M(t + \Delta t)\right] \tag{10}$$

where $B(t + \Delta t)$ is the magnetic flux density in [T] at the time step $t + \Delta t$ and $H(t + \Delta t)$ is the field strength at the next time step in [A/m].

## 2.2 Metaheuristic-Search Algorithms

In this study, the selected metaheuristic algorithms are GA, PSO, and DE. The parameters that these algorithms need to approximate are the JA parameters: $M_{\text{s}}$, $a$, $\alpha$, $c$ and $k$. At the beginning of the execution, depending on the initialization strategy, each algorithm creates a population of possible solutions. The following steps are individual for each algorithm, briefly explained as follows:

- **GA**: selects two candidate solutions (*parents*) from the population. These candidates undergo crossover and mutation to create a new chromosome (*child*). As the process is repeated, the algorithm produces better results [7].

- **PSO**: population consists of particles, each characterized by its position and velocity that evolves every iteration. Each particle remembers its best individual position and the best global solution (leader's position) over all iterations. The algorithm reaches convergence by sharing the information of each particle with the rest of the group (*swarm*) [13].

- **DE**: is based on generating a mutant for each individual, by combining three randomly selected individuals from the population. Each individual and its mutant are combined through a crossover operation to create a trial. If the trial solution outperforms the original individual, the trial solution takes the original position of the individual [14].

## 2.3 Parameter Initialization Analysis

Different strategies can be used to perform the initialization of the JA parameters The simplest technique is trial-and-error. However, depending on the search space and resolution, this algorithm can be imprecise and computationally inefficient. Most of the literature studies using metaheuristic algorithms for magnetic materials ignore the initialization phase and employ a random initialization strategy with default settings [15, 16]. Alternative solutions such as normal PDF parameter initializations have been shown to improve convergence and computational time by considering expert knowledge and transformer data [9]. However, the most limiting factor of the latest strategy is confidence in expert knowledge, which can be subjective and vary between experts. Moreover, the reliability of the employed data can also be a limiting factor. In order to model these concerns, this research work presents a framework to combine different perspectives and their uncertainties by using p-boxes. The parameter initialization methods mentioned above are explained as follows.

### 2.3.1 Uniform Probability Density Function

The uniform PDF parameter initialization is carried out by setting limits for each parameter and then randomly choosing a value inside these boundaries. Therefore, the probability of selecting any value is the same. The uniform parameter initialization is defined as:

$$f(x) = \begin{cases} \frac{1}{u-l} & \text{for } l \leq x \leq u, \\ 0 & \text{for } x < l \text{ or } x > u. \end{cases} \tag{11}$$

where $l$ and $u$ are lower and upper boundaries of each parameter, respectively.

### 2.3.2 Normal Probability Density Function

Depending on the available data and the level of expert knowledge, the normal PDF parameter initialization is configured with different mean and variance or uncertainty. In this case, the initialization should be modelled as follows:

$$f(x) = \frac{1}{\sigma\sqrt{2\pi}} e^{\left(-\frac{(x-\mu)^2}{2\sigma^2}\right)} \tag{12}$$

where $\mu$ is the mean value and $\sigma$ is the standard deviation.

### 2.3.3 Probability Boxes

P-boxes are structures created by combining probability theory and interval arithmetic. These structures allow for the propagation of aleatoric and epistemic uncertainty [10]. Probability boxes are defined by left and right bounds on the distribution function of a quantity and additional information

constraining (*i*) the mean and variance to specified intervals and (*ii*) the distributional shape. A p-box is a set of distribution functions $F$ satisfying the following constraints, for specified distribution functions $\underline{F}$, $\overline{F}$ and specified bounds $m_1 < m_2$ on the expected value of the distribution and specified bounds $v_1 \leq v_2$ on the variance of the distribution [10]:

$$
\underline{F}(x) \leq F(x) \leq \overline{F}(x)
$$

$$
m_1 \leq \int_{-\infty}^{\infty} x \, dF(x) \leq m_2
$$

$$
v_1 \leq \int_{-\infty}^{\infty} x^2 \, dF(x) - \left( \int_{-\infty}^{\infty} x^2 \, dF(x) \right)^2 \leq v_2
$$

(13)

P-boxes serve the same role for random variables that upper and lower probabilities serve for events. They can model uncertain decision-making scenarios when different sources of information are available or the confidence in the available information is unknown. Namely, different CDFs represent different pieces of knowledge of the process under consideration, with different uncertainties. Each CDF has a mean and a variance, just like a normal PDF. The construction of a p-box can be done by combining all the available information, modelled as normal PDFs. A simple example of this process can be seen in Figure 2, where in Figure 2a two different normal PDFs are illustrated and in Figure 2b, their combination into a p-box as CDFs is presented [17].

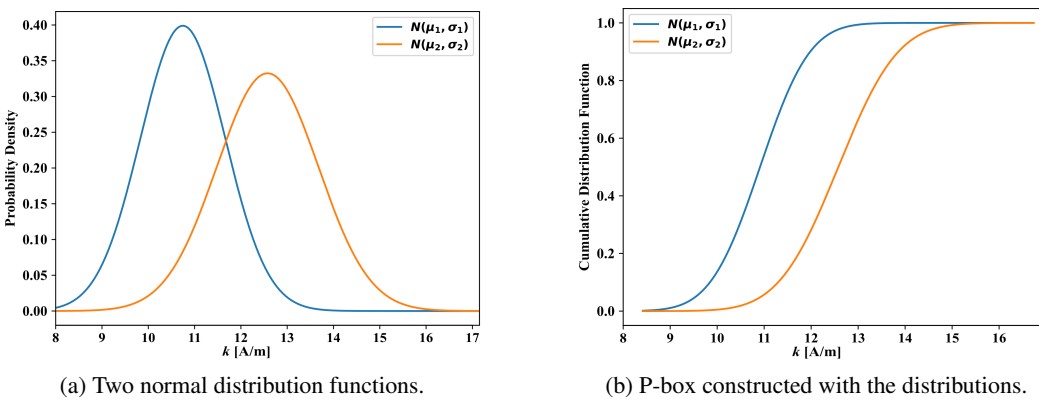

(a) Two normal distribution functions.

(b) P-box constructed with the distributions.

Figure 2: P-box construction example.

## 3 Proposed Approach

The proposed approach to estimate the JA parameters is shown in Figure 3.

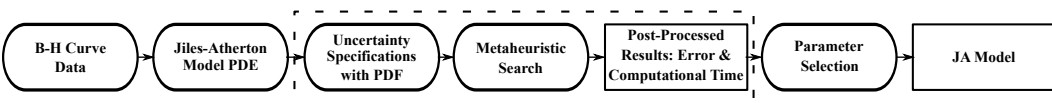

Figure 3: Overall block diagram of the framework based on [9].

First, in order to obtain the JA model, the $BH$ `hysteresis loop data` of the analysed material is needed. In this case, two different $BH$ curves are analyzed. Material A is a H75-23 core used in medium voltage transformers, see Figure 1, and material B has been obtained from an open-source database of $BH$ curves [18], see Figure 4.

The $BH$ curve data is used to obtain the `JA model PDE` (cf. Eq. (8)). The three stages covered by a dashed box in Figure 3 are part of the computational part. The execution starts with JA parameter initialization through different `uncertainty levels`, modelled with PDFs. The limits of the JA parameters are set based on available information. For material A, the saturation is known and $M_s$ is not included in the estimation procedure. The rest of material A parameters are set based on approximations and values in the literature [1, 7]. The limits for material B are higher because

previous research supports this hypothesis [18]. The chosen limits for both materials are presented in Table 1.

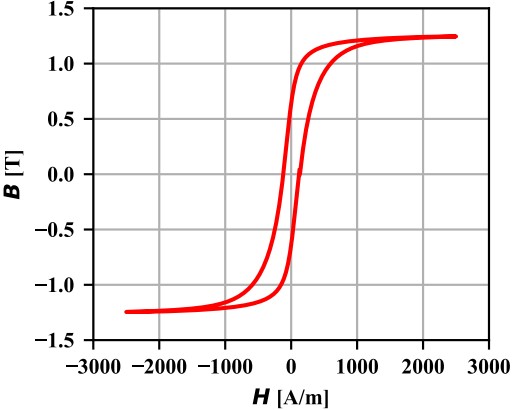

Figure 4: $BH$ curve of material B.

Table 1: Limits of Jiles-Atherton parameters for both materials.

| Parameter | Limits Material A | | Limits Material B | | Unit |
|:---:|:---:|:---:|:---:|:---:|:---:|
| | Lower | Upper | Lower | Upper | |
| $M_s$ | $1.52 \times 10^6$ | $1.52 \times 10^6$ | $9.35 \times 10^5$ | $11.42 \times 10^5$ | A/m |
| $a$ | 1 | 100 | 1 | 1000 | A/m |
| $\alpha$ | $10^{-6}$ | $10^{-5}$ | $10^{-6}$ | $10^{-5}$ | [-] |
| $c$ | 0.1 | 0.9 | 0.1 | 0.9 | [-] |
| $k$ | 1 | 100 | 1 | 5000 | A/m |

At this stage, the different parameter initialization strategies are analyzed and listed below.

- **Uniform PDF**: a random uniform distribution is used to initialize the JA parameters, denoted as $U(u, l)$. This case presents the scenario with the least knowledge about the material. The selection of suitable boundaries for each JA parameter is necessary (cf. Table 1).

- **Normal PDF**: the parameter initialization is performed with normal PDFs with three levels of uncertainty. The uncertainty levels are represented through standard deviations $\sigma = \{1\%, 5\%, 10\%\}$ which can be formalized as: $N(\mu, 0.01\mu)$, $N(\mu, 0.05\mu)$, and $N(\mu, 0.1\mu)$. Hence, three scenarios with different levels of knowledge are tested separately.

- **Probability-box**: the parameter initialization is performed by constructing a p-box for each parameter. These p-boxes are the combination of levels of knowledge modelled as normal PDFs. In this research, the p-boxes are formed by merging normal PDFs with $1\%$, $5\%$ and $10\%$ uncertainties, specified as in the normal distribution approach. An example of the combination of different normal CDFs for the $k$ parameter is shown in Figure 5a.

  The upper and lower limits of the p-box are shown in black in Figure 5b. The parameter initialization process is performed first by randomly selecting a value between 0 and 1. This value is translated into the p-box through inverse sampling [10] and the maximum and minimum boundaries are selected. Considering that each value between the selected boundaries has different probabilities, with the median having the highest value and the boundaries having the lowest, the initial parameter is sorted out by considering a normal distribution between the boundaries. Figure 5b shows this strategy with the example random value of 0.6 and the corresponding normal PDF with lower and upper limits at 12 and 13.2.

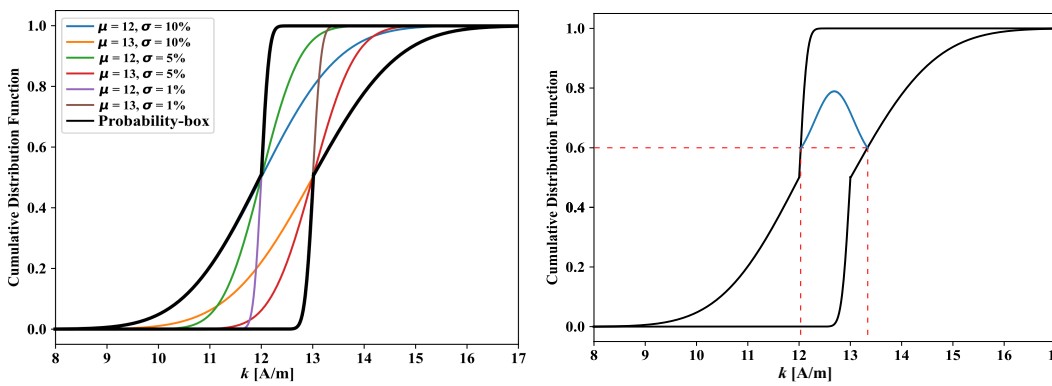

(a) Example of a p-box construction from different normal CDFs for JA parameter $k$.

(b) Illustration of parameter initialization selection with p-box strategy.

Figure 5: Comparison of p-box construction and parameter initialization strategy.

The parameter initialization is followed by executing the chosen `metaheuristic-search` algorithms: GA, PSO or DE. The `error` is calculated by the root mean square error $\varepsilon$ between the measured and calculated flux density in %:

$$\varepsilon = \frac{1}{B_{\mathrm{s}}}\sqrt{\frac{1}{N_B}\sum_{i=1}^{N_B}\left(B_{\mathrm{data}_i} - B_{\mathrm{cal}_i}\right)^2} \times 100 \qquad (14)$$

where $B_{\mathrm{s}}$ is the saturation value of the flux density in [T], $i$ is the current iteration, $N_B$ is the number of flux density values, $B_{\mathrm{data}_i}$ is the flux density value in [T] of the material at each iteration and $B_{\mathrm{cal}_i}$ is the estimated flux density in [T] of the material at each iteration. The error and computational time obtained in each iteration are stored for `post-processing`. After the execution stage is finished, the optimal `parameter selection` for each metaheuristic algorithm and parameter initialization strategy is selected and used to `tune the JA model`.

The algorithm ends when (*i*) it converges, *i.e.* the error has not changed in the last 100 iterations or (*ii*) a maximum number of iterations has been reached ($it_{\max}$ = 1000, in this study). Then, the error, computational time and JA optimized parameters are stored, and the same procedure is repeated $N_{\mathrm{it}}$ = 10000 times to obtain the error and computational time distributions. After $N_{\mathrm{it}}$ trials it is possible to compare different algorithms with statistically relevant results. After trying all the different parameter initialization and metaheuristic algorithm combinations, the strategies with the best trade-off between accuracy and computational time are compared.

## 4 Numerical Results

In this section, the best accuracy and computation time results are presented for the uniform PDF, normal PDF, and p-box parameter initialization strategies. A more detailed analysis of the results obtained with the first two parameter initialization approaches can be found in [9].

### 4.1 Material A

Figure 6 shows the comparison between accuracy and computational time among the best cases of each initialization strategy and Table 2 displays different statistics inferred from error and computational time distributions such as confidence interval (CI). For uniform and normal PDF parameter initializations the best results are obtained with DE and GA with 5% uncertainty, respectively. The best results for p-box parameter initialization strategy are obtained with the GA metaheuristic algorithm. The findings indicate that the p-box yields the highest accuracy with the highest probability of getting an error of 2.1%. However, the normal PDF parameter initialization strategy demonstrates a shorter computational time compared to the p-box. The normal PDF parameter initialization results show maximum likelihood of ending the computation after 1 second, while the p-box case is more likely to last 2 seconds or longer. This increased execution time for the p-box is justified by the inverse

sampling process needed to construct the p-box from different PDFs and the wider search space. Still, the computational time with p-box is lower than with uniform PDF parameter initialization, with a maximum likelihood of ending the computation in 6 seconds.

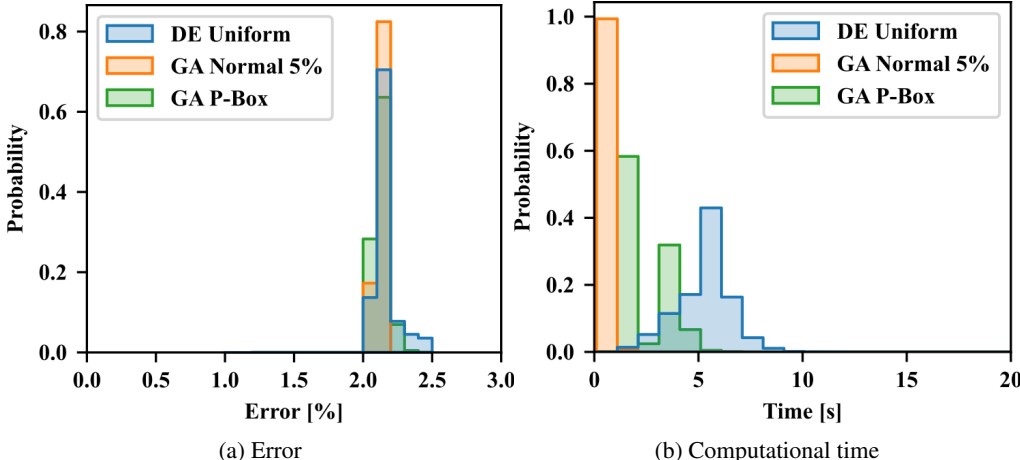

(a) Error              (b) Computational time

Figure 6: Error and computational time PDFs for the best metaheuristic algorithms with the different parameter initialization strategies for material A.

Note that the p-box computational time is acceptable, although it is not the shortest. In addition, the p-box approach provides greater flexibility for the specification of the initial parameters. In contrast, the normal PDF initialization strategy requires greater confidence in the knowledge of experts or available information.

Table 2: Summary of error and computational time results for material A (cf. Figure 6).

| Parameter | Case | Uniform DE | Normal GA 5% | P-box GA |
|---|---|---|---|---|
| Error [%] | Best/Worst | 2.1/2.5 | 2.1/2.2 | 2.1/2.5 |
| | Max. Likelihood | 2.2 | 2.1 | 2.1 |
| | 95% Upper/Lower | 2.1/2.4 | 2.1/2.1 | 2.1/2.2 |
| Time [s] | Best/Worst | 1/11 | <1/2 | 1/6 |
| | Max. Likelihood | 6 | 1 | 2 |
| | 95% CI Upper/Lower | 2/8 | <1/1 | 1/5 |

## 4.2 Material B

Figure 7 shows the best results obtained with each parameter initialization strategy for material B and Table 3 displays different statistics inferred from the error and computational time distributions obtained for material B. The best results are obtained with the DE algorithm for uniform parameter initialization and PSO with 1% uncertainty for normal PDF. For p-box parameter initialization, the best outcomes are obtained when the GA algorithm is used.

The findings reveal that the accuracy of the p-box strategy improves the uniform and normal PDF parameter initialization strategies, with the highest probability of getting a minimum error of 1.9% compared to the uniform and normal PDF parameter initializations with 1.7% and 1.8% errors, respectively. Even though the maximum likelihood error is higher for p-box, the best case accuracy is lower than for the rest of the parameter initialization strategies. Moreover, the computational time for the p-box parameter initialization strategy is longer than for normal PDF. For normal PDF parameter initialization, the computation has the maximum likelihood of lasting 2 seconds, while for the p-box parameter initialization, it is 3 seconds. In addition, the computational time for the p-box strategy is shorter than for the uniform PDF parameter initialization, with a maximum likelihood time of 5 seconds. In contrast, as shown in Table 3, the 95% upper bound CI is higher for p-box parameter initialization, 20 seconds, than for uniform PDF, 17 seconds. This difference can be attributed to

the broader parameter range considered by the p-box initialization approach. Considering all these results, the computational time for p-box parameter initialization can be considered acceptable.

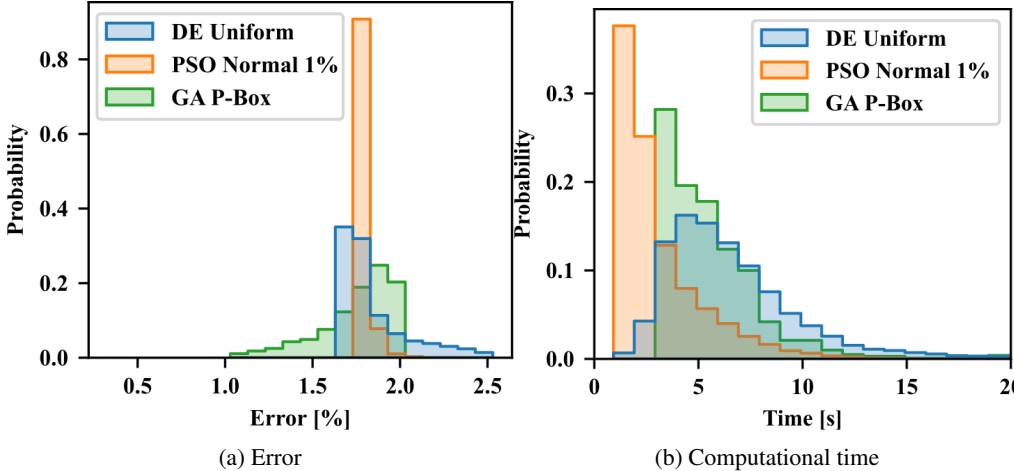

(a) Error

(b) Computational time

Figure 7: Error and computational time PDFs for different algorithms with the different parameter initialization strategies for material B.

Table 3: Summary of error and computational time results for material B (cf. Figure 7).

| Parameter | Case | Uniform DE | Normal PSO 1% | P-box GA |
|---|---|---|---|---|
| Error [%] | Best/Worst | 1.7/2.5 | 1.7/2.4 | 1.0/2.0 |
| | Max. Likelihood | 1.7 | 1.8 | 1.9 |
| | 95% Upper/Lower | 1.7/1.9 | 1.8/1.8 | 1.1/2 |
| Time [s] | Best/Worst | 1/41 | 1/23 | 3/43 |
| | Max. Likelihood | 5 | 2 | 3 |
| | 95% CI Upper/Lower | 3/17 | 1/9 | 3/20 |

## 5 Conclusions

This study presents a framework for evaluating the initialization of Jiles Atherton (JA) parameters using uniform, normal, and probability-box (p-box) distributions. Unlike uniform and normal distributions, p-boxes can combine different sources of material data, expert knowledge, and its uncertainty to accurately estimate the JA parameters. The proposed approach evaluates the effectiveness of p-box parameter initialization for two different materials by analyzing their propagation through metaheuristic algorithms: Genetic Algorithms, Particle Swarm Optimization, and Differential Evolution. The algorithm with the best accuracy and computational time results for each material is compared with parameter initialization strategies that have demonstrated satisfactory performance in JA parameter estimation: the classical random uniform distribution and normal probability density function (PDF).

Obtained results indicate that with p-boxes, the accuracy of the results is improved. However, the computational time is increased compared to the normal PDF parameter initialization. This increase is due to the inverse sampling process needed to construct the p-box from different PDFs and the broader parameter range that is analyzed with this strategy. Overall, p-box parameter initialization has resulted in a higher accuracy and longer computational time than uniform parameter initialization. Therefore, the p-box approach emerges as an alternative technique for JA parameter initialization and metaheuristics-based estimation when there are few material data and expert knowledge.

The proposed framework is based on the combination of different normal PDFs to construct a p-box. However, future research may explore the use of alternative PDFs for the generation of p-boxes, which could better represent expert knowledge about the magnetic material and the transformer.

## Acknowledgment

This research was funded by the Spanish State Research Agency (grant No. CPP2021-008580) and Department of Education of the Basque Government, Research Group Program (grant No. IT1634-22 and IT1504-22). In addition, Jose I. Aizpurua is funded by the Ramón y Cajal Fellowship, Spanish State Research Agency (grant number RYC2022-037300-I), co-funded by MCIU/AEI/10.13039/501100011033 and FSE+.

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
