# OpenReview forum: "Probability-Box Informed Hysteresis Modelling through Metaheuristic Search Algorithms"
_MAEB/2025/Congreso — MAEB 2025_

### Official Review · Reviewer_TiLa · 2025-03-10
**This work introduces a probability-box (p-box) initialization strategy within the framework of three different metaheuristics, namely GA, PSO and DE, for the parameter estimation of the Jiles-Atherton model for hysteresis modeling. Two different magnetic materials are used to evaluate the impact of the newly developed initialization. The idea of using p-boxes to integrate several sources of uncertainty is interesting.**

**Rating:** 4
**Confidence:** 4

**Review:**

This work introduces a probability-box (p-box) initialization strategy within the framework of three different metaheuristics, namely GA, PSO and DE, for the parameter estimation of the Jiles-Atherton model for hysteresis modeling. Two different magnetic materials are used to evaluate the impact of the newly developed initialization. The idea of using p-boxes to integrate several sources of uncertainty is interesting. The paper is also well written, but the following issues have raised concerning the experimentation performed:

1.- The main concern is that the experiments are hardly reproducible, as details about the operators, application rates, etc. of the algorithms used are not fully detailed.

2.- On a related matter, the presentation of the results is somehow confusing as the authors redirect the reader to [9], where a more in-depth analysis is presented. However, the full data is  required to be included in the paper so as to make it self-contained.

3.- .- The authors have used a combined stopping condition, which makes it difficult to evaluate the results properly. Indeed, if any of the algorithm is misconfigured so that it gets stuck quickly in a local minimum, the runtime would be much shorter. Data about the hit-rate is therefore required. That is, how many runs reach the predefined maximum number of function evaluations.

4.- As stated above, Figures 6 and 7, and Tables 2 and 3 only display data from small set of configurations. As to the actual data, it is suggested to add more precision to the numerical data, as few differences can be identified.

5.- Even the p-box strategy is innovative, it reached marginal results. Not sure if assuming a gaussian distribution between the boundaries of the p-box is a good strategy. The authors may want to evaluate additional PDFs.

---

### Official Review · Reviewer_VBfe · 2025-03-13
**Probability-Box Informed Hysteresis Modelling through Metaheuristic Search Algorithms**

**Rating:** 5
**Confidence:** 3

**Review:**

The present paper sets out the findings of an experimental evaluation of three methods to estimate the five parameters of the Jiles-Atherton model. The purpose was to reproduce transient phenomena on magnetic materials. The paper is well structured, presenting a fundamental theory and background, as well as a well-conducted experimental evaluation.

Although the field of application of the metaheuristics falls outside the scope of my own research expertise, I believe that this paper merits acceptance on account of the relevance of the domain, the meticulously designed experimental evaluation, and the results obtained, which provide a robust substantiation of the conclusions.

---

### Official Review · Reviewer_d8pr · 2025-03-16
**Novel initialisation strategy based on p-boxes combined with GA, DE and PSO to determine the parameters of a JA model**

**Rating:** 4
**Confidence:** 4

**Review:**

This paper applies three different metaheuristics (genetic algorithm, differential evolution and particle swarm optimisation) to estimate the JA parameters of a hysteresis model. The paper's originality lies in the initialisation stage, where the usage of p-boxes are proposed by the authors. This initialisation method is compared to a random initialisation and an initialisation procedure based on Probability Density Functions. Two distinct iron core materials are selected for testing.

The paper is well-written and organised. However, there are three key concerns that need to be addressed: firstly, the actual contribution of the paper (is it an initialisation strategy?), secondly, the soundness of the experimental evaluation, and thirdly, the reproducibility of the results.

The paper could be significantly improved by addressing the following points:
- More details should be given about GA, DE and PSO, as well as their parameterisations.
- The experimental evaluation could be improved by designing and explaining it better. For instance, the number of repetitions of the runs performed for each approach should be specified. It is also essential to apply a statistical comparison procedure to the results obtained by the different solvers. The above would provide statistical substantiation for the conclusions drawn.

Pros:
- The real-world application is interesting.

Cons:
- Originality
- Technical quality
- Reproducibility

---

### Decision · Program_Chairs · 2025-03-19

Accept